# Dermal Absorption of Sesquiterpene Lactones from Arnica Tincture

**DOI:** 10.3390/pharmaceutics14040742

**Published:** 2022-03-29

**Authors:** Franziska M. Jürgens, Fabian C. Herrmann, Sara M. Robledo, Thomas J. Schmidt

**Affiliations:** 1Institute of Pharmaceutical Biology and Phytochemistry, University of Münster, PharmaCampus, Corrensstrasse 48, D-48149 Münster, Germany; franziska.juergens@uni-muenster.de (F.M.J.); fabian.herrmann@uni-muenster.de (F.C.H.); 2PECET-School of Medicine, University of Antioquia, Calle 70 # 52-21, Medellin 0500100, Colombia; sara.robledo@udea.edu.co

**Keywords:** Arnica tincture, sesquiterpene lactones, helenalin, natural products, dermal absorption, skin penetration, diffusion cells, fluorescence microscopy

## Abstract

Arnica tincture is a traditional herbal medicine used to treat blunt injuries, e.g., bruises and squeezes. In addition, a potential new use in the treatment of cutaneous leishmaniasis is currently under investigation. Therefore, detailed information about the dermal absorption of the tincture and especially its bioactive constituents, sesquiterpene lactones (STLs) of the helenalin- and 11α,13-dihydrohelenalin type, is mandatory. Consequently, this article reports on dermal absorption studies of Arnica tincture using diffusion cells and porcine skin as well as two human skin samples with different permeability. The amounts of STLs on the skin surfaces, in skin extracts and in the receptor fluids were quantified by ultra-high-performance liquid chromatography with high-resolution mass spectrometry (UHPLC-HRMS). It was found that Arnica STLs permeated into the receptor fluid already 4 h after the application, but the amount was rather low. Within 48 h, a maximum of 8.4%, 14.6% and 36.4% of STLs permeated through porcine skin, human skin A (trans-epidermal water loss (TEWL) = 11.518 g·m^−2^·h^−1^) and the more permeable human skin B (TEWL = 17.271 g·m^−2^·h^−1^), respectively. The majority of STLs was absorbed (penetrated into the skin; 97.6%, 97.8% and 99.3%) after 48 h but a huge portion could not be extracted from skin and is expected to be irreversibly bound to skin proteins. To better visualize the analytes in different skin layers, a fluorescence-labeled STL, helenalin 3,4-dimethoxycinnamate, was synthesized. Fluorescence microscopic images depict an accumulation of the fluorescent derivative in the epidermis. For the treatment of local, cutaneous complaints, an enrichment of the bioactive substances in the skin may be considered beneficial.

## 1. Introduction

Arnica tincture, prepared by extraction with 70% ethanol from the flowerheads of *Arnica montana* L., Asteraceae, is a traditionally used herbal medicine for the topical treatment of injuries and inflammations. The tincture contains a variety of constituents, e.g., STLs, flavonoids and phenolic acids [1]. The STLs helenalin, 11α,13-dihydrohelenalin, and their carboxylic acid esters are known to be the main bioactive substances in Arnica preparations. They have numerous biological activities, e.g., anti-inflammatory, antitumoral, antibacterial, antifungal and antiarthritic [1]. Furthermore, antileishmanial activity of Arnica tincture and isolated Arnica STLs has been reported [2,3]. In our recent in vivo experiment with experimental cutaneous leishmaniasis (CL) in golden hamsters, Arnica tincture showed comparable or even better curative effects than the standard drug glucantime, thus a new field of application of Arnica tincture was discussed [4].

Some pharmacological products applied topically to the skin can penetrate to a certain extent into the cutaneous layers and exert some effect there. This is the case with topical formulations for the treatment of cutaneous inflammatory diseases such as CL. Although Arnica tincture has a long traditional use as a topical drug, detailed data on the dermal absorption are not available so far. Some studies have investigated the permeation of Arnica formulations through pig ear skin into ethanol/water (4/6, *v*/*v*) [5] or the permeation of STLs through silastic membranes and human epidermis into methanol/water (1/1, *v*/*v*) [6]. In another investigation, the permeation enhancers dimethyl sulfoxide, oleic acid, lauroglycol, isopropyl myristate, or Tween 80 were used as receptor fluids [7]. A different skin absorption method, i.e., stripping skin with adhesive tape, was used to study the penetration of Arnica STLs into different layers of the *stratum corneum* [8]. However, none of these studies on dermal absorption of Arnica STLs completely fulfilled the organization for economic cooperation and development (OECD) requirements.

In particular, the OECD guideline for dermal absorption of chemicals demands the use of human or animal skin, diffusion cells and physiological receptor fluids maintained at normal skin temperature, i.e., 32 ± 1 °C [9]. Minipigs were classified as the animal model of choice because of similar permeation characteristics compared with human skin, but nevertheless the use of human skin is considered most relevant [10,11,12]. In the case of pigs, skin from the flanks, back or ears is suitable; in case of humans, breast or abdominal skin is commonly used [11]. Skin permeability can be evaluated by the trans-epidermal water loss. Since Göttingen minipigs are bred in such a way that there is little interindividual variation [13], TEWL does not need to be determined each time for this species, and normal TEWL values of healthy minipig skin are up to 10–15 g·m^−2^·h^−1^ [14]. For human skin, stronger interindividual variation is common [10] and a cutoff TEWL value of 10 g·m^−2^·h^−1^ has proven suitable to differentiate between intact and altered human skin [15]. In the case of CL, a higher skin permeability resulting in higher TEWL values is typical. In *Leishmania* infected skin, twice as high TEWL values were measured compared to uninfected skin [16].

In this work, skin absorption experiments were carried out that, for the first time, comply with the OECD guideline on skin absorption. In particular, diffusion cell experiments were performed with a physiological receptor fluid and the suggested porcine and human skin samples at the normal skin temperature.

## 2. Materials and Methods

### 2.1. Chemicals and Materials

All chemicals were purchased from Merck KGaA (Darmstadt, Germany) or Thermo Fisher Scientific (Schwerte, Germany) if not stated otherwise. α-Santonin was acquired from Carl Roth GmbH & Co. KG (Karlsruhe, Germany). Water was deionized by a Synergy water purification system (Merck KGaA, Darmstadt, Germany). Phosphate buffered saline (PBS) pellets were obtained from Th. Geyer GmbH & Co. KG (Renningen, Germany) and dissolved in deionized water to obtain 1 × PBS with pH 7.4. Arnica tincture from the manufacturer Hetterich (batch number 144801) was purchased in a local pharmacy. Oasis HLB SPE cartridges were purchased from Waters Corporation (Milford, MA, USA). Dry dichloromethane, 1-ethyl-3-(3-dimethylaminopropyl) carbodiimide and 4-dimethyl amino pyridine were provided by Bernhard Wünsch (Institute of Pharmaceutical and Medicinal Chemistry, University of Münster, Germany). Helenalin was isolated in previous projects with the method described in [17] and a purity of >95%. Göttingen minipig skin from the flank (internal donor ID 347400) and human abdomen skin (internal donor ID 1732, TEWL = 11.518 ± 2.534 g·m^−2^·h^−1^ and internal donor ID 1728, TEWL = 17.271 ± 0.570 g·m^−2^·h^−1^, removed during clinical surgery) was provided by LTS Lohmann Therapie Systeme AG (Andernach, Germany). TEWL values were determined in triplicates at different locations on the same piece of tissue. Ethical review and approval were not required for this study because only ex vivo materials were used that came from plastic surgery with the patient’s consent or directly from the breeder (Ellegaard Göttingen Minipigs A/S, Dalmose, Denmark).

### 2.2. Synthesis of Helenalin 3,4-Dimethoxycinnamate (H-DMCA)

3,4-Dimethoxycinnamic acid (DMCA, 15.6 mg, 0.08 mmol), helenalin (21.9 mg, 0.08 mmol), 1-ethyl-3-(3-dimethylaminopropyl) carbodiimide hydrochloride (14.4 mg, 0.08 mmol) and 4-dimethyl amino pyridine (18.4 mg, 0.15 mmol) were dissolved in dry dichloromethane (5 mL) and stirred at room temperature. Loss of alcohol was monitored by thin layer chromatography (silica gel plates 60 F_254_, Merck KGaA, Darmstadt, Germany with mobile phase ethyl acetate/hexane (4/1)). After 24 h, the solvent was evaporated under reduced pressure. The residue was dissolved in acetonitrile and purified by preparative HPLC (Jasco, Groß-Umstadt, Germany) with preparative reverse phase column Reprosil 100 C-18 (5 μm, 250 mm, 20 mm, Macherey-Nagel, Düren, Germany). The yield was 14.97 mg. NMR spectra were recorded on an Agilent DD2 600 MHz spectrometer and calibrated with the solvent (CDCl_3_) peak as reference. NMR data are reported in Appendix A. The purity as determined by UHPLC with diode array detector (DAD) was >95% (Appendix A).

### 2.3. Dermal Absorption Experiments

Porcine and human skin were already prepared (shaved and fat removed) when provided by LTS Lohmann Therapie Systeme AG. The samples were dermatomized to a thickness of 600 µm (Göttingen minipig skin) or 500 µm (human skin) to include epidermis and the upper part of dermis. Skin punches of 4.6 cm^2^ (Arnica tincture) and 5.7 cm^2^ (H-DMCA) were prepared with a hollow punch. Layer thickness was checked with a coating thickness gauge (APT Surfix N, PHYNIX Sensortechnik GmbH, Neuss, Germany). For each experiment, six skin punches from the same donor skin portion were used. Permeation experiments were performed based on the OECD guideline [9]: skin samples were fixed between donor and acceptor chambers of diffusion cells. Together with a magnetic stirring bar, the receptor fluid (PBS, pH 7.4) was introduced into the acceptor chamber so that skin samples were completely wetted. The experiments with Arnica tincture were performed with in-house custom-made equable diffusion cells of LTS Lohmann Therapie Systeme AG with Microette T (Hanson research, Chatsworth, CA, USA) as donor compartment (V = 4.5 mL) and acceptor compartments with V = 10 mL. Experiments with H-DMCA were performed with different in-house custom-made diffusion cells (D1-D7) with different acceptor chamber volumes of 7.6 mL (D1), 7.8 mL (D2), 8.1 mL (D3), 8.9 mL (D4), 8.0 mL (D5), 8.0 mL (D6) and 8.0 mL (D7) and donor chamber volumes of 6.9 mL (D1), 7.3 mL (D2), 7.0 mL (D3), 6.9 mL (D4), 5.1 mL (D5), 6.1 mL (D6) and 7.0 mL (D7). The test substance (500 µL of Arnica tincture or 200 µL of H-DMCA) was applied on the skin surface. Blanks were carried along without treatment or with 70% ethanol, respectively. The diffusion cells were placed in a water bath on a magnetic stirring plate and incubated at 32 ± 1 °C and 700 rpm. After 4 h, 8 h, 24 h and 32 h, the receptor fluid was removed in its entirety and replaced with the same amount of prewarmed, new PBS (32 ± 1 °C, pH 7.4). After 48 h, the experiment was terminated, and the receptor fluid was collected. Further, the donor chamber and the upper side of the skin were rinsed with 2 mL of PBS (pH 7.4) while the skin was still fixed in the diffusion cell. Skin samples were then removed, cut into small pieces and extracted with methanol for 24 h at 1500 rpm on a laboratory shaker. The internal standard α-santonin was added to skin extracts, receptor fluids and skin wash solutions. Receptor fluids and skin wash solutions were desalinated with Oasis HLB SPE cartridges. Last, solvents were evaporated, and samples were dissolved in water/acetonitrile (95/5, *v*/*v*) to suit UHPLC-HRMS starting conditions.

### 2.4. UHPLC-HRMS Analysis

UHPLC-HRMS analysis was performed with an Ultimate 3000 UHPLC system (Dionex, Sunnyvale, CA, USA) coupled to a micrOTOF-Q II quadrupole-time-of-flight mass spectrometer (Bruker Daltonics, Billerica, MA, USA). For chromatographic separation, an Acclaim RSLC 120 C18 analytical column (particle size 2.2 μm, diameter 2.1 mm, length 0.1 m; Dionex, Sunnyvale, CA, USA) at 40 °C and a flow rate of 0.4 mL/min were used. Due to different STL concentrations, the injection volume varied from 2 µL (high concentrated skin wash solutions) to 20 µL (skin extracts and receptor fluids) and to 200 µL (analysis of low concentrated metabolites). The binary gradient of water (A) and acetonitrile (B) with 0.1% formic acid each (*v*/*v*/*v*/*v*) started at 18% B, increased to 100% B within 9.0 min, was held at 100% B until 15.0 min, returned to 18% B until 15.1 min and stayed at 18% B until 20.0 min. The eluate of the first minute was not passed to the mass detector to wash away salts. As nebulizer gas (3.5 bar), dry gas (9 L/min, 200 °C) and collision gas nitrogen was used. Electrospray ionization (ESI) was performed in positive ionization mode with a capillary voltage of +4500 V. Full transition experiments were carried out with a mass range of *m*/*z* 50–1500. MS/MS analysis was carried out with a collision energy of 20 eV. For mass calibration, sodium formate (10 mM) in isopropanol/water/formic acid/sodium hydroxide (50/50/0.2/1, *v*/*v*) was used. Data acquisition, operation of the system and data processing were carried out with the software packages Chromeleon (Dionex, Sunnyvale, CA, USA) and OTOF control (Bruker Daltonics, Billerica, MA, USA). For the analysis of metabolic activity, data processing was performed with the software Compass DataAnalysis 4.1 (Bruker Daltonics, Billerica, MA, USA). In particular, the dissect compounds algorithm was used with a S/N threshold of 1, a maximum number of five overlapping compounds, a mass spectrum cutoff intensity of 0.1% and an expected chromatographic peak width (full width at half maximum) of ≥3 spectra. The generated dissect compounds were then examined for expected metabolites. Furthermore, the intensity (I) of metabolite peaks was evaluated in the base peak chromatograms to differentiate between low (I < 1000), medium (1000 < I < 10,000) and high (I > 10,000) relative metabolite formation.

### 2.5. UHPLC-HRMS Method Validation

The limit of detection (LOD, S/N > 3) was determined to be 0.3 ng/mL and the lowest limit of quantification (LLOQ, S/N ≥ 10) 1.0 ng/mL for all analytes. For the quantification, α-santonin, an STL not occurring in Arnica, was used as the internal standard (IS). To compensate different detector response of the analytes compared with the IS, substance specific correction factors were determined. The UHPLC-HRMS method was validated by applying the Guideline on Bioanalytical Method Validation M10 of the International Council for Harmonisation of Technical Requirements for Pharmaceuticals for Human Use (ICH) [18]. All criteria for linearity, accuracy, precision, selectivity, specificity, carry-over and matrix effects were fulfilled (see Appendix A). Furthermore, the stability of analytes after three freeze-thaw cycles (freeze-thaw stability), 8 h at room temperature (bench top stability), 24 h at 10 °C in the UHPLC-HRMS autosampler (autosampler stability), twelve months at −20 °C (long term stability) and 48 h at 40 °C was confirmed. Last, an influence of sample filtration and dilution was excluded.

### 2.6. UHPLC-FLD Analysis

Fluorescence analysis was carried out on an Acquity UPLC H-class with Acquity FLR detector and Waters Empower 3 Software (Waters, Milford, MA, USA). Analytical column Acquity UPLC HSST3 (particle size 1.8 μm, diameter 2.1 mm, length 100 mm) at 40 °C and a binary gradient of water (A) and acetonitrile (B) with 0.1% formic acid each (*v*/*v*) was used (0.0 min 2% B, 8.0 min 100% B, 9.0 min 100% B, 9.1 min 2% B, 10 min 2% B). Injection volume was set to 2 μL and flow rate to 0.5 mL/min. Fluorescence detection (FLD) was carried out with excitation at 327 nm and emission at 390–490 nm.

### 2.7. Microscopy

Skin samples were embedded in polyethylene glycol (PEG 4000) employing deionized water as intermedium (PEG/H_2_O dilution series 15/85, 25/75, 50/50, 70/30, 95/5 and two final steps of absolute and H_2_O-free PEG, each infunded at 64 °C for 1 h under gentle agitation). The skin samples were subsequently oriented in embedding molds, covered with freshly molten absolute PEG 4000 and solidified at room temperature. Sectioning was carried out on a custom-made sledge microtome, yielding serial transverse skin sections of around 50 µm in thickness. The sections were afterwards transferred into deionized water to remove the embedding medium completely before mounting the specimens on standard glass slides. Light and epifluorescence microscopy was performed employing a Leitz/Leica Orthoplan microscope equipped with semi-apochromatic optics (PL Fluotar 10/0.30, PL Fluotar 25/0.60 and PL Fluotar 63/0.90), a 200 W HBO light source and a dichroitic filter cube for DAPI.

## 3. Results and Discussion

### 3.1. Characterization of Arnica Tincture

Prior to the application of Arnica tincture on skin samples, an UHPLC-HRMS analysis was performed to characterize the tincture under study. In particular, the STL pattern and quantity in the tincture were of interest. To estimate the dermal absorption of the Arnica STLs, their molecular weight (MW), number of hydrogen bond acceptors (HBA), number of hydrogen bond donors (HBD) and (calculated) octanol-water partition coefficients (*c*logP) were compared with the recommended values by Choy and Prausnitz, who extrapolated Lipinski’s Rule of Five on dermal instead of oral absorption. They recommend stricter thresholds of *c*logP (0.0–5.0), MW (<335 Da), HBA (≤5) and HBD (≤2) [19]. In the present Arnica tincture, helenalin (H), 11α,13-dihydrohelenalin (DH), and its esters, i.e., acetate (ac), methacrylate (ma), isobutyrate (ib), tiglinate (t) and 2- and 3-methyl butyrate (mb), were identified. Table 1 gives their quantified levels and molecular properties that are relevant for dermal absorption. The Arnica STLs fulfill the enhanced criteria of *c*logP, HBA and HBD, but four STLs (Ht, Hmb, DHt, DHmb) slightly exceed the new limit of MW. Nevertheless, no change in absorption of these four STLs could be observed in our experiments (Section 3.2).

The European Pharmacopoeia (Ph. Eur.) requires a minimum total content of 0.04% STLs in Arnica tinctures and the use of *Arnica montana* flowerheads of the Central European type [20]. In the Arnica tincture under study a total STL amount of 524.00 ± 24.27 µg/mL (=0.05%, *m*/*v*) was quantified, and the STL pattern shows a predominant amount of H derivatives compared to the DH derivatives (Table 1 and Figure 1), which is a characteristic of Central European *Arnica montana* flowerheads [21]. Consequently, it can be confirmed that the tincture under study fulfills the Ph. Eur. criteria.

### 3.2. Dermal Absorption of Arnica Tincture

For dermal absorption studies, skin of Göttingen minipigs and two different human skin samples were used. In order to obtain insights into the permeation of STLs through intact skin and through altered skin, a human skin with slightly increased permeability (human skin A, TEWL = 11.518 g·m^−2^·h^−1^) and a human skin with high permeability (human skin B, TEWL = 17.271 g·m^−2^·h^−1^) were chosen in addition to the skin of healthy minipigs. Arnica tincture was applied on six equable samples of porcine skin and both human skins in diffusion cells, and one blank sample without treatment was carried out in each case. The physiological receptor fluid was sampled after defined time intervals. At the end of the experiment, the skin surface and donor chamber were rinsed (=skin wash solution), and the skin samples were cut and extracted (=skin extract). Subsequently, STLs were quantified in the receptor fluids, skin extracts and skin wash solutions (Figure 1) by UHPLC-HRMS. In ESI mass spectra, H and its derivatives display fragmentation to a common “core” fragment at *m*/*z* 245.1172, whereas DH and its derivatives form the corresponding core fragment at *m*/*z* 247.1329. Therefore, extracted ion chromatograms (EIC) of *m*/*z* 245.1172 and *m*/*z* 247.1329 were used for the evaluation of the STLs. A small portion of the applied STLs remained on the skin surface and was recovered in the skin wash solution but the majority of all STLs penetrated into the skin and through the skin, into the receptor fluid.

It is striking that, compared to the applied Arnica tincture, the H/DH ratio has changed in favor of DHs in the skin wash solution, the skin extract and in the receptor fluid. It is probable that penetration of Hs into the skin was stronger than that of DHs, resulting in fewer Hs on the skin surface. In contrast, DHs were slightly better extractable from the skin and showed slightly stronger permeation through the skin. This finding agrees with a result from Wagner et al. who found a better permeation/extractability of DHac compared to Hib [8]. They considered the different *c*logP values responsible for that, with DHac having a lower *c*logP value than Hib. In comparison of the *c*logP values of all Arnica STLs (Table 1), no big differences can be observed between Hs and corresponding DHs. Similarly, the other molecular properties that affect dermal absorption (Table 1) are very similar for Hs and corresponding DHs. Consequently, no explanation for the different absorption behavior can be drawn based on these properties.

Regarding H and DH core structures, the only structural difference is the C-11/C-13 methylene group of Hs, which is hydrogenated in case of DHs. This methylene group is part of a Michael acceptor in addition to the cyclopentenone moiety. It is known that STLs react with sulfhydryl groups of cysteine via conjugations at the Michael acceptors [22]. Besides the conjugation with free cysteine, this reaction was also observed with peptides and proteins [22,23,24]. Therefore, a reaction with cysteines of skin proteins is considered highly likely and should be higher in the case of Hs with two Michael acceptor systems than in case of DHs with only one Michael acceptor system.

In Figure 1, the chromatograms of porcine samples are shown exemplarily. Penetration into the skin, permeation through the skin and residual non-absorbed amounts on the skin were also observed with the human skin samples. Table 2 gives an overview of the STL levels in receptor fluids, skin extracts and skin wash solutions. Of the applied 0.5 mL Arnica tincture (i.e., 262 µg STLs), residual amounts of only 2.4%, 2.2% and 0.7% were found in the skin wash solutions of porcine skin, human skin A and human skin B, respectively. Since the amount of substance remaining in the skin is considered as absorbed (according to the OECD guideline on skin absorption [9]), this indicates that 97.6%, 97.8% and 99.3% of the applied dosage was absorbed. The percentages of absorbed analytes are rather high and comparable between the human skins with different permeability and between porcine and human skin. Somewhat bigger differences were seen in the amounts that permeated into the receptor fluids (8.4%, 14.6% and 36.4%).

With the STL amounts that were not absorbed (skin wash solution) and the amounts that reached the receptor fluids quantified, the residual amounts must remain in the skin. Interestingly, only 6.9%, 7.8% and 6.0% could be extracted from cut porcine and human skin samples A and B, respectively. The formation of irreversible bonds with skin proteins as assumed above might be an explanation for this observation. Strong STL interaction with proteins was also observed for human serum albumin, to which approx. 40% of STLs from Arnica tincture of the Central European type were bond [23]. For this effect, contributions of cysteine binding, reactions with other amino acids and non-covalent interactions were discussed [23].

The strongest inter- and intraspecies differences were seen in the STL amount that reached the receptor fluids. In Figure 2, the permeation profiles are shown in terms of cumulative permeated absolute amounts of the analytes per skin area (left ordinate axis) and the percent of applied dose (right ordinate axis). Steady-state flux is determined from the slope of the linear parts of the graphs and describes permeation through a unit area of skin in a unit of time [11].

After 4 h, the first samples were taken from the receptor fluids, and for porcine skin and human skin A and B, low percentages of the STLs were already quantified (0.1%, 0.1% and 0.2%). After another 4 h, further 0.2%, 0.1% and 2.5% permeated into the receptor fluids. At this point, a difference could already be seen between human skin A and B. After 24 h, a cumulative amount of 1.6%, 2.7% and 16.3% permeated through the skin. After 32 h, this percentage increased to 2.9%, 6.4% and 27.4%. Finally, 48 h after the application, a total percentage of 8.4%, 14.6% and 36.4% of the STLs permeated through porcine skin, human skin A and human skin B, respectively. In accordance, the steady-state flux through human skin B (0.54 µg·cm^−2^·h^−1^) was considerably higher than through human skin A (0.28 µg·cm^−2^·h^−1^) or porcine skin (0.17 µg·cm^−2^·h^−1^).

It is noticeable that the majority of permeated STLs (corresponding to 81%, 82% and 55% of total permeated STLs) reached the receptor fluid within the second 24 h (24–48 h). The permeation profiles show the typical lag phase, followed by a linear phase. This can be explained by the three steps of dermal absorption: penetration (entry into a skin layer, i.e., tissue accumulation), permeation (penetration through skin layers) and resorption (uptake into the circulatory system or receptor fluid) [25]. A substance is already considered absorbed after penetration into the outer skin layer, whereas it is considered resorbed only with the uptake into the receptor fluid. In the lag phase, penetration prevails. Next, in the linear phase, permeation and resorption are constant, resulting in a steady state flux. The lag phase can be determined by the x-axis intercept of the regression line. For porcine skin and human skin A, the lag phase was seen within the first 20 h and 19 h, respectively. Then, the linear phase was observed. In contrast to this, the permeation profile of human skin B with higher permeability showed the transition from lag phase to linear phase already after 5 h. Additionally, a flattening of the permeation curves can be recognized after 48 h, which implies that absorption approached completeness. In contrast to this, resorption into the receptor fluid was probably not finished yet in case of porcine skin and human skin A.

In comparison of the three different skins used in our experiments, in general similar permeation characteristics were seen for porcine skin and human skin A. This agrees with the high comparability of human and porcine skin [26]. Nevertheless, testing with human skin provides valuable information and highlights high interindividual differences in this experiment. Human skin B compared to human skin A had substantially stronger permeability (factor 2.5), which could already be expected based on the higher TEWL value. However, even with this high permeability, the major fraction of STLs did not reach the receptor fluids and remained in the skin instead.

Our results show similarities with the previous literature on dermal absorption of Arnica STLs [6,7]. A generally low permeation of Arnica STLs without dimethyl sulfoxide or oleic acid supplement is consistent with the findings of Bergonzi et al. [7]. In the experiments of Tekko et al., only two DH derivatives of the STL mixture in the applied Arnica tincture were detected in the receptor fluid [6]. This agrees with our observation that resorption of DH derivatives is more comprehensive than resorption of H derivatives (see Appendix A).

### 3.3. Metabolic Activity of Skin Samples

Fresh skin provides metabolic activity due to drug metabolizing enzymes such as cytochrome P450 (CYP) and *N*-acetyltransferases (NAT) [27]. Dermal hydroxylation by CYP enzymes is known, e.g., for all-*trans* retinoic acid and cholecalciferol [27,28]. *N*-acetylation in human skin was reported for, e.g., the allergen paraphenylenediamine [29]. In our recent in vitro metabolism experiments with pig and human liver microsomes, Arnica STLs were extensively metabolized to glutathione conjugates, cysteine conjugates, hydroxides, hydrates and combinations thereof [17]. Therefore, skin extracts and receptor fluids were analyzed for dermal STL metabolites and hydrates, *N-*acetylcysteine conjugates (NAC), cysteine conjugates, hydroxides and hydroxylated hydrates were detected (Table 3).

Hydrates and NAC were identified for both Hs and DHs in the receptor fluids of human and pig samples, but interestingly, no H-NAC were present in the skin extracts. Moreover, only DH hydroxides as well as hydroxylated DH hydrates were formed, which are, therefore, probably hydroxylated at C-11 of DHs, which is not accessible for hydroxylation in case of H derivatives. Instead, small amounts of cysteine conjugates were only found in case of H derivatives. It is probable that cysteine is added to the exocyclic methylene group of Hs (C-11/C-13), which is hydrogenated in case of DHs. The observed dermal metabolism of Hs differs from the previously studied hepatic metabolism: hydrates were the major H metabolites in dermal metabolism, whereas glutathione conjugation prevailed in hepatic metabolism. Nevertheless, small amounts of H cysteine conjugates and NAC (“mercapturic acids”) of both STL series could be identified, which likely derive from previous glutathione conjugation. It is possible that a portion of Hs is bound to cysteine in skin proteins and is hence not available for glutathione conjugation and other metabolic reactions. In contrast to this, dermal metabolism of DHs is consistent with the hepatic metabolism. In both biotransformations, hydroxides, hydrates and hydroxylated hydrates were among the major DH metabolites. Hydroxylation of DHs was extensive and resulted in an even higher abundance of DH hydroxides compared to native STLs in the receptor fluids after 48 h (Figure 3). The extensive formation of metabolites contributes to the explanation of why a large fraction of the STLs was not detected in skin extracts, skin wash solutions and receptor fluids.

### 3.4. Synthesis and Characterization of a Fluorescence-Labeled Sesquiterpene Lactone

To examine the interaction of STLs with different skin layers, a fluorescence-labeled STL was planned to be synthesized. In order to compare the dermal absorption behaviors, the fluorescent STL should have a similar MW and lipophilicity compared to native Arnica STLs. Arnica STLs showed a similar dermal absorption behavior without influence of the ester group. Instead, absorption appeared to be influenced more by the physicochemical properties of the common H or DH backbones, and a stronger retention of H esters in skin was apparent. Consequently, a fluorescent H ester was chosen to be synthesized. Since the MW of the native esterified carboxylic acids varies from 60 g/mol (acetic acid) to 102 g/mol (methylbutanoic acid), only a very small fluorescent molecule could be used. Cinnamic acid (148 g/mol) and its derivatives have low MWs, and the carboxylic acid group enables an esterification with the hydroxy group of helenalin. The smallest cinnamic acid derivative emitting light in the visible spectral region appeared to be 3,4-dimethoxycinnamic acid (DMCA), which was chosen as fluorescence label. The helenalin 3,4-dimethoxycinnamic acid ester (H-DMCA) was synthesized by Steglich esterification, and its properties (MW = 452.18 g/mol; *c*logP = 3.3), even though somewhat higher, are still comparable to the native Arnica STL esters (MW = 304–348 g/mol, *c*logP = 1.5–2.9).

Analysis of the reaction mixture by UHPLC-HRMS revealed the formation of H-DMCA (70.4%) in addition to residual amounts of intermediates and a common urea byproduct of Steglich esterification [30]. After purification by preparative HPLC, the fluorescence-labeled STL was available in sufficient amount and purity for the skin penetration study. Since no fluorescent helenalin derivative has been described before, a characterization of the new fluorescent substance was performed. Spectroscopic, spectrometric and nuclear magnetic resonance (NMR) analysis are shown in the Appendix A and confirmed the identity and purity of the product. H-DMCA possesses three absorption maxima, i.e., λ = 227 nm, 236 nm and 327 nm. Excitation at the latter led to maximal fluorescence with emission at λ = 425 nm in UHPLC-FLD analysis. Exact mass was confirmed (5 ppm deviation of measured and calculated protonated molecule). For structure confirmation, ^1^H- and ^13^C-NMR spectra and 2D-NMR spectra (COSY, HSQC and HMBC) were evaluated. Full ^1^H- and ^13^C-NMR data is reported in Appendix A.

### 3.5. Dermal Absorption Experiments with the Fluorescence-Labeled Sesquiterpene Lactone

To examine dermal absorption of H-DMCA, the diffusion cell method and Göttingen minipig skin (600 µm thickness, 5.7 cm^2^) were used. To ensure sufficient fluorescence yield in the subsequent microscopic analysis, 2 mg of H-DMCA were applied on each skin sample. Since Arnica tincture is prepared with 70% ethanol (*v*/*v*), the analyte was dissolved in the same solvent, and a control skin was treated with the solvent only. After 4 h, 8 h, 24 h, 32 h and 48 h, samples from the receptor fluid were collected and analyzed for H-DMCA quantity. In Figure 4, the cumulative permeated amounts per square centimeter skin and the percent of the applied dosage are shown.

After 8 h, small amounts (0.1%) of H-DMCA were quantified in the receptor fluid. This amount remained very low with cumulative amounts of 0.3% after 24 h, 0.6% after 32 h and 0.9% after 48 h. The typical lag phase was observable within the first 5 h. Then, the linear phase with steady-state flux of 0.06 ng·cm^−2^·h^−1^ followed. Based on the observation that the last sampling point is included in the linear phase, it is possible that the permeation was not completed after 48 h. Nevertheless, the total resorbed amount after 48 h is quite low and below the resorbed amount of native Arnica STLs with porcine skin (8.4%). Less permeation of single STLs in ethanolic solution compared to STLs in Arnica tincture was also reported by previous authors, who used the skin stripping method with adhesive tape [8]. This observation might be due to the natural occurrence of penetration enhancers such as monoterpenes in the tincture [8] and could also explain the higher amount of resorbed Arnica STLs in comparison with H-DMCA, although the different molecular properties of the latter may also be relevant.

Further, skin washes and skin extracts were analyzed for H-DMCA quantity (Table 4). In skin washes, 14.7% of the applied dosage was recovered. Consequently, a total amount of 85.3% was absorbed. This is less than the absorbed portion of Arnica STLs (97.7%) but still the majority of the applied substance.

In skin extracts, 70.1% of the analyte were quantified, leading to a total recovery of 85.2%. The missing 14.8% probably remained in the skin as discussed for the native Arnica STLs. However, the extractable amount of H-DMCA from skin samples was much higher compared to Arnica STLs (6.9%). Consequently, the absorption characteristics of H-DMCA show differences (less resorption into the receptor fluid, enhanced extractability from skin) and similarities with the native Arnica STLs (extensive absorption into the skin, unextractable portion probably remaining in the skin). Therefore, it can be concluded that the fluorescent STL might also interact with skin proteins but to a lower extent compared to native Arnica STLs.

### 3.6. Microscopic Analysis of Skin Penetration

For the evaluation of skin penetration and the distribution of the fluorescent analyte in different skin layers, fluorescence microscopy was carried out. Cross sections through PEG embedded skin samples showed blue fluorescence of H-DMCA over the complete outer skin area (Figure 5), which was not observable in the control. The minipig skin samples under study comprise epidermis (50–65 µm thickness) and dermis (1200–2300 µm thickness, increasing with age) [10]. The latter consists of connective tissues composed of collagen and elastin fibers [31] in addition to appendages such as sebaceous glands on hair follicles and sweat glands in its upper part [32]. A hair fragment in the control skin (brighter structure in panel C2) shows intrinsic fluorescence, which was exceeded in the skin sample after treatment with the fluorescent STL (Figure 5B2). This observation might be explained by cysteine rich hair composition of keratin and keratin associated proteins with up to >30% cysteine [33] to which STLs might bind. In addition, accumulation of the lipophilic analyte in sebaceous secretion surrounding the hair is conceivable.

In contrast to this, the collagen and elastin fibers of the dermis show a slight intrinsic luminescence (Figure 5C [34]) which was not increased by treatment with H-DMCA (Figure 5A,B). Consequently, STL accumulation does not take place in the dermis but in the epidermis where the intensity of fluorescence is concentrated, as obvious in panels A1-B2 of Figure 5. To identify the affected epidermis layers, higher magnification was used and brightfield images were taken (Figure 6). The epidermis comprises a viable hydrophilic part and a non-viable lipophilic part, the *stratum corneum* (10 µm thickness) [10]. The latter is composed of dead keratinocytes and a lipid matrix and is further divided into the *stratum corneum conjunctum* with dense cells and the *stratum corneum disjunctum* with loosely attached, desquamating cells [35].

Obviously, most of the fluorescence is seen in the *stratum corneum*. The microscopic images even show the differentiation into the desquamating *stratum corneum disjunctum* and the tighter *stratum corneum conjunctum*. In addition to the bright fluorescence in the *stratum corneum*, a slight fluorescence is seen in the underlying hydrophilic epidermis layers (Figure 6A1,A2). The accumulation of the analyte in the epidermis, especially the *stratum corneum*, might result from interactions with cysteine rich keratin and might be supported by its lipophilicity (*c*logP = 3.3) that enables accumulation in the lipid matrix. However, if the analyte had only accumulated in the lipid matrix, it would have been removed completely from the skin by extraction with methanol. Since a significant portion could not be extracted from the skin, irreversible interactions with skin proteins are very likely. Covalent modification of proteins has been reported for STLs. Conjugation with cysteine sulfhydryl groups is expected but other interactions cannot be excluded. Drug accumulation in the skin might be beneficial for topical CL drugs because thereby the drug may be enriched at the site of infection. Further, fewer unwanted effects are expected when the drug is not absorbed to a high extent into the circulatory system [16].

## 4. Conclusions

In this article, dermal absorption experiments of Arnica tincture applied on porcine and human skin (“ex vivo”) in diffusion cells under near physiological conditions are reported. The absorption characteristics of the bioactive constituents, the sesquiterpene lactones, are studied for the first time in detail and according to the OECD guideline on skin absorption. Arnica sesquiterpene lactones were absorbed extensively into porcine and human skin without being influenced by skin permeability. Low to moderate amounts permeated through the skin samples. Comparable results were obtained with unaltered porcine and human skin. Further, metabolic activity in the skin led to the formation of (*N*-acetylated) cysteine-conjugates, hydrates and hydroxides, that were detected in both skin extracts and receptor fluids. The low amount of STLs resorbed into the system and the observed metabolism make it unlikely that dermal application of Arnica tincture might be of toxicological concern. Interestingly, the permeation and resorption of 11α,13-dihydrohelenalin derivatives was considerably higher compared to the helenalin derivatives. This hints at an interaction of the sesquiterpene lactones with skin proteins via their reactive Michael acceptors. Indeed, a significant portion of sesquiterpene lactones neither permeated nor could it be extracted from the skin. This portion is expected to be bound to skin proteins. Analogous experiments with a fluorescent semi-synthetic STL derivative (helenalin 3,4-dimethoxycinnamate) showed an accumulation of the STL in the epidermis. For the local treatment of cutaneous leishmaniasis lesions, and also for the traditional use of Arnica tincture, accumulation of the bioactive substances at the site of infection or injury (as compared to systemic uptake) may be considered advantageous.

## Figures and Tables

**Figure 1 pharmaceutics-14-00742-f001:**
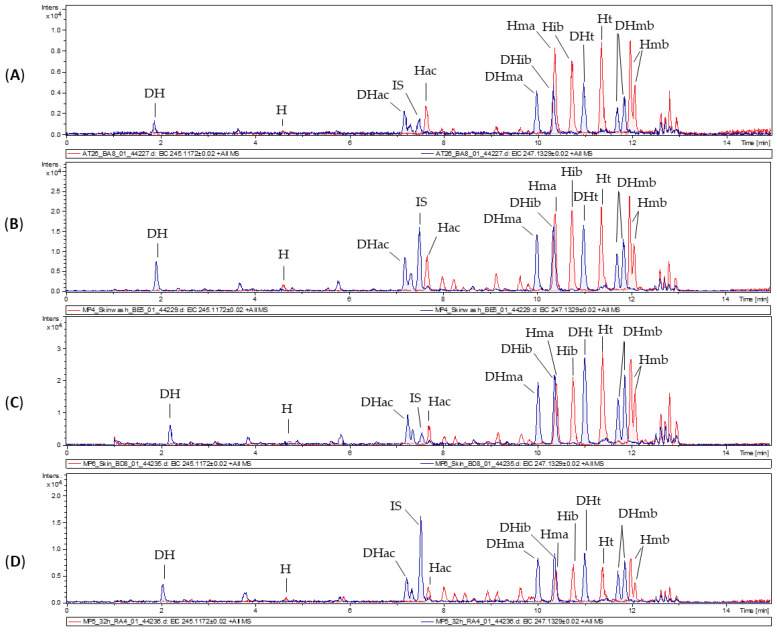
UHPLC-HRMS chromatograms with EICs of *m*/*z* 245.1172 (red, Hs) and *m*/*z* 247.1329 (blue, DHs) of the applied Arnica tincture (**A**) and representative samples of a receptor fluid (**B**), a skin extract (**C**) and a skin wash solution (**D**).

**Figure 2 pharmaceutics-14-00742-f002:**
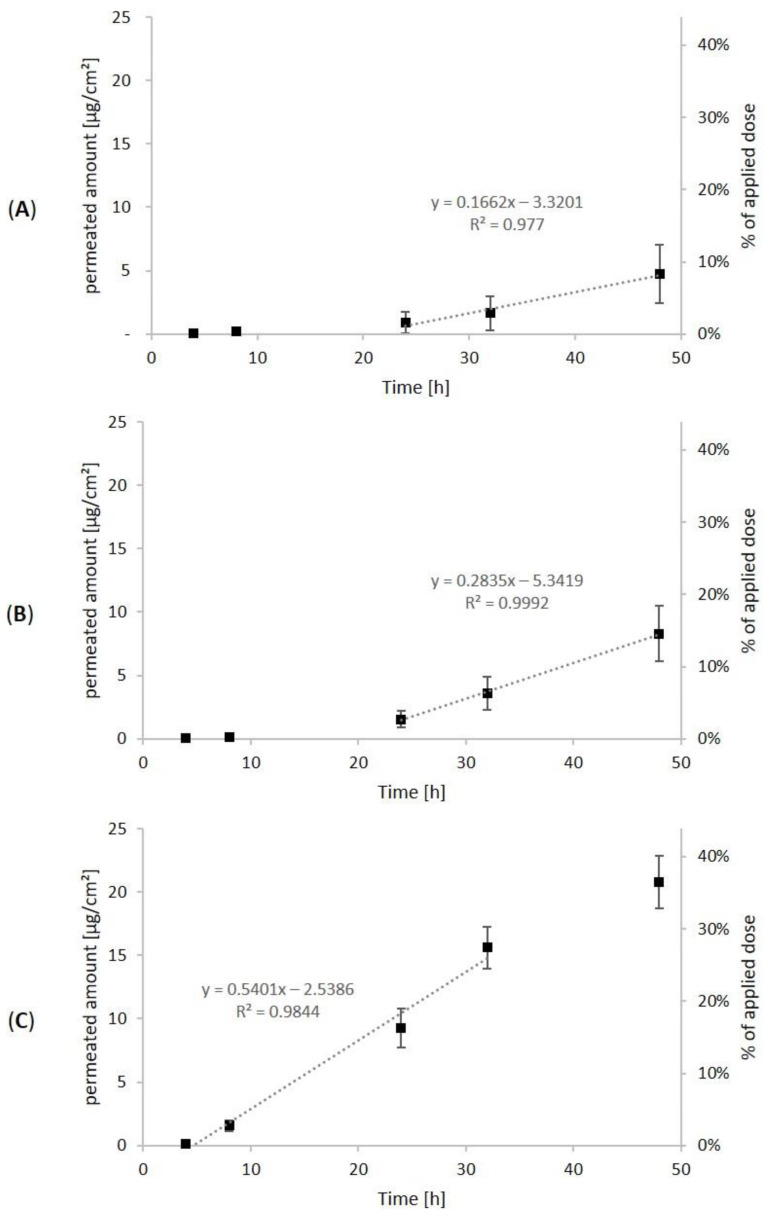
Cumulative permeated STL amount relative to skin area (left axis) and percent of total applied dose (right axis) in the receptor fluid over time (4 h to 48 h) after application of 0.5 mL Arnica tincture on porcine skin (**A**), human skin A (**B**) and human skin B (**C**), *n* = 6. The dashed regression line is based on the absolute permeated amounts (left axis). It represents the linear range used to calculate the steady-state flux, which is the slope of the regression line.

**Figure 3 pharmaceutics-14-00742-f003:**
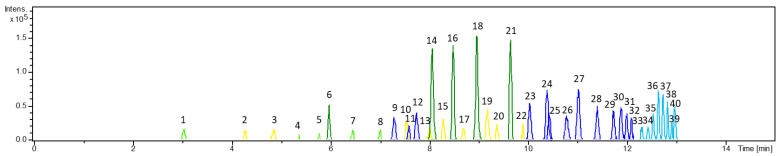
UHPLC-HRMS analysis of a receptor fluid after 48 h with dissect compound chromatograms of DH hydroxides (dark green, 6: DHac-OH, 14: DHma-OH, 16: DHib-OH, 18: DHt-OH, 21: DHmb-OH), unmetabolized STLs (dark blue, 9: DHac, 11: IS, 12: Hac, 23: DHma, 24: DHib, 25: Hma, 26: Hib, 27: DHt, 28: Ht, 29–30: DHmb, 31–32: Hmb), STL ethanol (EtOH) adducts (light blue, 33: Hac-EtOH, 34: DHma-EtOH, 35: DHib-EtOH, 36: Hma-EtOH, 37: Hib-EtOH, 38: Ht-EtOH, 39: DHmb-EtOH, 40: Hmb-EtOH), hydroxylated DH hydrates (light green, 1: DHac-OH-H_2_O, 4: DHma-OH-H_2_O, 5: DHib-OH-H_2,_O, 7: DHt-OH-H_2_O, 8: DHmb-OH-H_2_O) and STL hydrates (yellow, 2: DHac-H_2_O, 3: Hac-H_2_O, 10: DHib-H_2_O, 13: Hma-H_2_O, 15: Hib-H_2_O, 17: DHt-H_2_O, 19: Ht-H_2_O, 20: DHmb-H_2_O, 22: Hmb-H_2_O).

**Figure 4 pharmaceutics-14-00742-f004:**
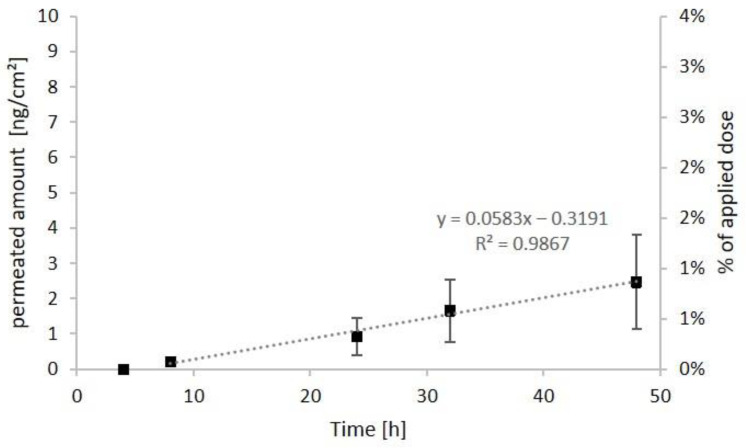
Cumulative permeated amount of H-DMCA relative to skin area (left axis) and percent of total applied dose (right axis) in the receptor fluid over time (4–48 h), *n* = 6. The dashed regression line is based on the absolute permeated amounts (left axis). It represents the linear range used to calculate the steady-state flux, which is the slope of the regression line.

**Figure 5 pharmaceutics-14-00742-f005:**
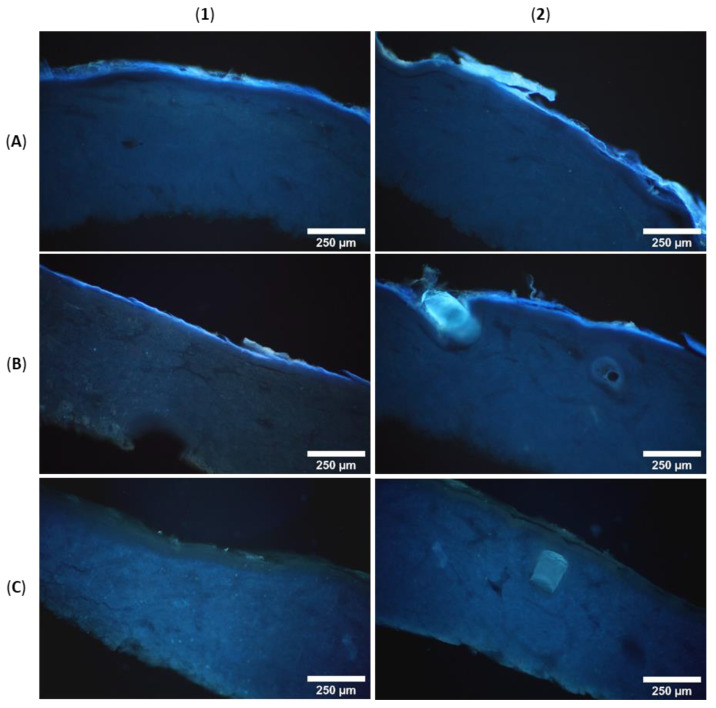
Fluorescence microscopic images of minipig skin after treatment with H-DMCA (**A**,**B**) and control skin (**C**).

**Figure 6 pharmaceutics-14-00742-f006:**
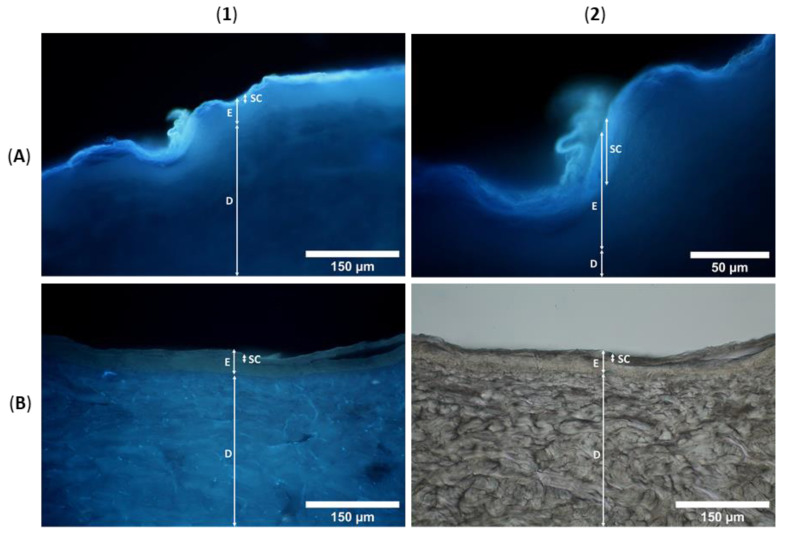
Fluorescence and brightfield microscopic images of minipig skin after treatment with H-DMCA (**A**) and control skin (**B**), SC = *stratum corneum*, E = epidermis, D = dermis.

**Table 1 pharmaceutics-14-00742-t001:** STLs in the Arnica tincture under study: formula, MW, HBA, HBD, *c*logP values and concentrations quantified by UHPLC-HRMS (*n* = 3).

STL	Formula	MW [g/mol]	HBA	HBD	*c*logP ^1^	c [µg/mL]
H	C_15_H_18_O_4_	262.3050	4	1	0.6513	15.26 ± 1.09
Hac	C_17_H_20_O_5_	304.3420	5	0	1.5038	24.79 ± 1.45
Hma	C_19_H_22_O_5_	330.3800	5	0	2.1470	69.22 ± 0.93
Hib	C_19_H_24_O_5_	332.3960	5	0	2.3418	51.50 ± 2.54
Ht	C_20_H_24_O_5_	344.4070	5	0	2.6760	76.68 ± 4.09
Hmb	C_20_H_26_O_5_	346.4230	5	0	2.8708	108.06 ± 4.23
DH	C_15_H_20_O_4_	264.3210	4	1	0.6781	65.50 ± 3.12
DHac	C_17_H_22_O_5_	306.3580	5	0	1.5305	11.24 ± 0.51
DHma	C_19_H_24_O_5_	332.3960	5	0	2.1738	20.19 ± 1.22
DHib	C_19_H_26_O_5_	334.4120	5	0	2.3685	46.66 ± 4.80
DHt	C_20_H_26_O_5_	346.4230	5	0	2.7028	12.31 ± 0.67
DHmb	C_20_H_28_O_5_	348.4390	5	0	2.8975	22.60 ± 0.59

^1^ calculated with ChemDraw Professional 20.1.

**Table 2 pharmaceutics-14-00742-t002:** STL portions in receptor fluids, skin extracts, skin wash solutions and remaining portions at the end of the dermal absorption experiments (48 h) with porcine skin, human skin A and human skin B (*n* = 6).

STL Percentage	Porcine Skin	Human Skin A	Human Skin B
receptor fluid	8.4 ± 4.0%	14.6 ± 3.8%	36.4 ± 3.6%
skin extract	6.9 ± 1.5%	7.8 ± 1.6%	6.0 ± 1.2%
skin wash solution	2.4 ± 0.7%	2.2 ± 0.7%	0.7 ± 0.3%
remaining in skin	82.3 ± 4.7%	75.4 ± 4.2%	56.8 ± 3.9%

**Table 3 pharmaceutics-14-00742-t003:** Metabolites in receptor fluids after 48 h (R), and skin extracts (S) of human skin A (HA), human skin B (HB) and pig skin (P). Differentiation in low (+, I < 1000), medium (++, 1000 < I < 10,000) and high (+++, I > 10,000) intensity or not detected (-) in base peak chromatograms.

Metabolite	Rt (min)	Formula	[M+H]^+^	RHA	RHB	RP	SHA	SHB	SP
Hac-H_2_O	4.9	C_15_H_20_O_5_	323.1490	++	++	++	++	++	++
Hma-H_2_O	8.0	C_19_H_24_O_6_	349.1646	++	++	++	++	++	++
Hib-H_2_O	8.2	C_19_H_26_O_6_	351.1803	++	++	++	++	++	++
Ht-H_2_O	9.1	C_20_H_26_O_6_	363.1803	++	++	++	++	+++	+++
Hmb-H_2_O	9.9	C_20_H_28_O_6_	365.1959	++	+++	++	++	++	++
DHac-H_2_O	4.3	C_15_H_22_O_5_	325.1646	+	++	+	++	++	++
DHma-H_2_O	7.0	C_19_H_26_O_6_	351.1803	++	++	++	++	++	++
DHib-H_2_O	7.4	C_19_H_28_O_6_	353.1959	++	++	++	++	++	++
DHt-H_2_O	8.7	C_20_H_28_O_6_	365.1959	++	++	++	++	++	++
DHmb-H_2_O	9.3	C_20_H_30_O_6_	367.2116	++	++	++	++	++	++
Hac-NAC	6.0	C_22_H_29_NO_8_S	468.1687	+	+	+	-	-	-
Hma-NAC	8.2	C_24_H_31_NO_8_S	494.1843	+	+	+	-	-	-
Hib-NAC	8.4	C_24_H_33_NO_8_S	496.1200	+	+	+	-	-	-
Ht-NAC	8.9	C_25_H_37_NO_8_S	508.2000	+	+	+	-	-	-
Hmb-NAC	9.4	C_25_H_35_NO_8_S	510.2156	+	+	+	-	-	-
DHma-NAC	7.8	C_24_H_33_NO_8_S	496.2000	+	+	+	++	++	++
DHib-NAC	8.1	C_24_H_35_NO_8_S	498.2156	+	+	+	++	++	++
DHt-NAC	8.6	C_25_H_39_NO_8_S	510.2156	+	+	+	++	++	++
DHmb-NAC	9.1	C_25_H_37_NO_8_S	512.2313	+	+	+	++	++	++
Hac-Cys	3.3	C_20_H_27_NO_7_S	426.1581	+	-	+	-	-	+
Hma-Cys	5.5	C_22_H_29_NO_7_S	452.1738	+	+	+	+	-	+
Hib-Cys	5.8	C_22_H_31_NO_7_S	454.1894	+	+	+	+	-	+
Ht-Cys	6.4	C_23_H_31_NO_7_S	466.1894	+	+	+	+	-	+
Hmb-Cys	6.8	C_23_H_33_NO_7_S	468.2051	+	+	+	+	-	+
DHac-OH	6.0	C_17_H_22_O_6_	323.1490	+++	+++	+++	++	++	++
DHma-OH	8.1	C_19_H_24_O_6_	349.1647	+++	+++	+++	+++	+++	+++
DHib-OH	8.5	C_19_H_26_O_6_	351.1803	+++	+++	+++	+++	+++	+++
DHt-OH	9.0	C_20_H_26_O_6_	363.1803	+++	+++	+++	+++	+++	+++
DHmb-OH	9.6	C_20_H_28_O_6_	365.1960	+++	+++	+++	+++	+++	+++
DHac-OH-H_2_O	3.0	C_17_H_24_O_7_	341.1595	++	++	++	+	+	+
DHma-OH-H_2_O	5.4	C_19_H_26_O_7_	367.1752	++	++	++	+	+	+
DHib-OH-H_2_O	5.8	C_19_H_28_O_7_	369.1908	++	++	++	+	+	+
DHt-OH-H_2_O	6.5	C_20_H_29_O_7_	381.1908	++	++	++	+	+	+
DHmb-OH-H_2_O	7.0	C_20_H_30_O_7_	383.2065	++	++	++	+	+	+

**Table 4 pharmaceutics-14-00742-t004:** Portions of H-DMCA in receptor fluids, skin extracts, skin washes and remaining portions in dermal absorption experiments with porcine skin (*n* = 6).

STL Percentage	Porcine Skin
receptor fluid	0.9 ± 0.2%
skin extract	70.1 ± 5.8%
skin wash	14.7 ± 6.5%
remaining in skin	14.4 ± 11.2%

## Data Availability

Data reported in this study is contained within the article. The underlying raw data is available on request from the corresponding author. The raw data are not publicly available due to the complexity and amount of data which requires special software for processing.

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
