# Peer review of "Dermal Absorption of Sesquiterpene Lactones from Arnica Tincture"

_pharmaceutics, 2022, doi:10.3390/pharmaceutics14040742_

Round 1
Reviewer 1 Report
This paper titled “Dermal Absorption of Sesquiterpene Lactones from Arnica Tincture” is interesting. This manuscript could be considered for publication in Pharmaceutics after major revising.
My comments are as follow:
- Check the abbreviations, the text format of full text, especially figures.
- There are some grammatical errors in the text.
- The security needs to be considered.
- The purity of H-DMCA should be checked.
- Ethics should be explained.
- Please compare your data with previous studies in the result and discussion section.
Author Response
Reviewer 1
- Check the abbreviations, the text format of full text, especially figures.
We checked the text format and adjusted the figure format. We inserted the abbreviation STL in the abstract (see comment of Reviewer 2) and defined the abbreviations UHPLC-HRMS in the abstract, DAD in Section 2.2, ICH in Section 2.5 and FLD in Section 2.6.
- There are some grammatical errors in the text.
We carefully checked for grammatical errors.
- The security needs to be considered.
We added a consideration regarding the toxicological potential of the absorbed test substances in Lines 528-530. Lab safety, safety of the used materials, etc. was also considered at all stages of the presented work.
- The purity of H-DMCA should be checked.
The purity of H-DMCA was checked by UHPLC-DAD analysis and information was added in Lines 111-112.
- Ethics should be explained.
Ethical review and approval were not required for this study because only ex vivo materials were used that came from plastic surgery with the patient’s consent or directly from the breeder (Ellegaard Göttingen Minipigs A/S, Dalmose, Denmark). We added this information in the Materials Section.
- Please compare your data with previous studies in the result and discussion section.
We did compare our data with those of previous studies wherever possible and cited these studies (references [6-8], [17], [22-24] and [26-29]).
We thank the reviewer for the efforts to help us improve our manuscript.
Reviewer 2 Report
The manuscript entitled “Dermal Absorption of Sesquiterpene Lactones from Arnica Tincture” by Franziska M. Jürgens et al. aims to evaluate the dermal absorption of the STLs contained in the Arnica tincture according to the organization for economic cooperation and development (OECD) guidelines. The manuscript appears relevant, the concept is interesting, and the scientific language is properly used. Furthermore, some minor issues should be addresses:
ABSTRACT:
- Lines 15, 18, 24: please avoid the use of the personal form.
- please insert the TEWL values for the employed skin samples
- line 22: does “absorbed into the skin” mean “accumulated into the tissue”? Generally the absorption process refers to active entry into the bloodstream and thus is should be better to talk about tissue accumulation. Moreover, it is not clear the timepoint at which the “absorption” data are related and these results sound strange together with the previously reported percentage of permeated actives.
- I suggest to insert the acronym of sesquiterpene lactones (STLs) also in the abstract to avoid long repetition and make it more fluid.
MATERIALS AND METHODS:
- please specify the number of repetition for the permeation assays.
- Line 89-91: Authors should better specify what they are ID 347400. ID 1732 and ID 1728. Are they ethical approval codes?
- line 111: the name and manufacturer of the coating thickness gauge is missing, please add it.
- Line 112: the type (e.g. Franz diffusion cell), characteristics (eg. the volume of both receptor and donor chamber), name and manufacturer of diffusion cell are missing, please add them.
- Lines 116-120: the volume both of the receptor chamber and the samples collected at each time points is missing, please improve this part.
- Line 126: this sentence is confused, what the “5% acetonitrile” is referred to? Seems to be related to the solvent, but is pure acetonitrile or a mixture with other solvents?
RESULTS AND DISCUSSION:
- lines 213-214: TEWL values are reported without a SE. Moreover, as the experiments should be repeated it is unlikely to imagine that different skin samples have identical TEWL. Please clarify if the repetition were carried out by using different portion of the same tissue or different similar tissues.
- Table 1: the calibration curves of all the respective STLs compound are missing, I suggest to add them in the supplementary material.
- lines 216-220: this is a repetition of what previously described in the methods section.
- Table 2: at which timepoint does the table refer? 48 h? please specify.
- Lines 225-227: this sentence is misleading, please improve it.
- Line 322: I strongly suggest adding the description of this paragraph in the respective methods section.
-figure 4 (1) the ordinate label is wrong, because the values are expressed as a percentage. I guess the authors intended the percentage of the dose .... This needs to be clarified. Also the ordinate label of figure 4 (1) is wrong… maybe it should be ng/cm2
- lines 412-414 There is a conceptual error in the presentation of the results. The flux is obtained from the slope of the straight line passing through the points ... the numerical data that the authors report are not related to the flux but of permeated amount per cm2 after a certain time interval. Authors need to reconsider their deductions
ALL ALONG THE MANUSCRIPT:
- please avoid the use of the personal form.
Author Response
Reviewer 2
ABSTRACT:
- Lines 15, 18, 24: please avoid the use of the personal form.
The personal form is no longer used.
- please insert the TEWL values for the employed skin samples
We inserted the TEWL values.
- line 22: does “absorbed into the skin” mean “accumulated into the tissue”? Generally the absorption process refers to active entry into the bloodstream and thus is should be better to talk about tissue accumulation. Moreover, it is not clear the timepoint at which the “absorption” data are related and these results sound strange together with the previously reported percentage of permeated actives.
Information on the timepoint (48 h) was added. In the OECD guideline it is written that “the test substance remaining in the skin should be considered as absorbed“. We also defined the different steps of absorption in Lines 320-324. For clarity in the abstract, the term “absorbed into the skin” was changed to “absorbed (penetrated into the skin)” (Line 24) and in Line 321 “tissue accumulation” was added for the skin penetration step.
- I suggest to insert the acronym of sesquiterpene lactones (STLs) also in the abstract to avoid long repetition and make it more fluid.
The acronym of sesquiterpene lactones (STLs) was inserted in the abstract.
MATERIALS AND METHODS:
- please specify the number of repetition for the permeation assays.
The number of used skin punches (technical replicates) was added.
- Line 89-91: Authors should better specify what they are ID 347400. ID 1732 and ID 1728. Are they ethical approval codes?
The IDs are internal donor IDs. We changed the description to “internal donor ID” for clarity. Ethical review and approval were not required for this study because only ex vivo materials were used that came from plastic surgery with the patient’s consent or directly from the breeder (Ellegaard Göttingen Minipigs A/S, Dalmose, Denmark). We added this information in the Materials Section.
- line 111: the name and manufacturer of the coating thickness gauge is missing, please add it.
The name and manufacturer of the coating thickness gauge were added.
- Line 112: the type (e.g. Franz diffusion cell), characteristics (eg. the volume of both receptor and donor chamber), name and manufacturer of diffusion cell are missing, please add them.
The volume of donor and acceptor chambers was added. The diffusion cells were in house custom-made, so no name and manufacturer can be provided. This information was added in the text.
- Lines 116-120: the volume both of the receptor chamber and the samples collected at each time points is missing, please improve this part.
At each time point the receptor fluid was removed in its entirety and replaced with new PBS so that skin samples were completely wetted. Because of different acceptor chamber sizes, the volumes differed slightly but were considered in the calculations of the results.
- Line 126: this sentence is confused, what the “5% acetonitrile” is referred to? Seems to be related to the solvent, but is pure acetonitrile or a mixture with other solvents?
“5% Acetonitrile” is referred the solvent which is acetonitrile in aqueous solution (5/95, v/v). We added (aqueous) to clarify the solvent mixture.
RESULTS AND DISCUSSION:
- lines 213-214: TEWL values are reported without a SE. Moreover, as the experiments should be repeated it is unlikely to imagine that different skin samples have identical TEWL. Please clarify if the repetition were carried out by using different portion of the same tissue or different similar tissues.
TEWL values were determined in triplicates at different locations on the same piece of tissue. This information and the deviations were added in the Materials Section.
- Table 1: the calibration curves of all the respective STLs compound are missing, I suggest to add them in the supplementary material.
The calibration curves were added in the Supplementary Materials (Figures S4-S11).
- lines 216-220: this is a repetition of what previously described in the methods section.
The redundant technical information was removed from these sentences. However, we feel that it is necessary to keep the main information as a service for the reader.
- Table 2: at which timepoint does the table refer? 48 h? please specify.
The timepoint was added in the Table caption.
- Lines 225-227: this sentence is misleading, please improve it.
We improved the sentence by specifying the distribution of STLs.
- Line 322: I strongly suggest adding the description of this paragraph in the respective methods section.
A description of this paragraph was added in the respective Methods Section 2.4.
-figure 4 (1) the ordinate label is wrong, because the values are expressed as a percentage. I guess the authors intended the percentage of the dose .... This needs to be clarified. Also the ordinate label of figure 4 (1) is wrong… maybe it should be ng/cm2
In the Figure captions of Figures 2 and 4 “percentage of total applied dose” was added. The ordinate label of Figures 2 (2) and 4 (2) are µg·cm-2·h-1 (Arnica STLs) and ng·cm-2·h-1 (H-DMCA) because in the OECD guidance document (reference [11]) flux is defined as “mass of test substance passing through a unit area of skin per unit of time under steady-state conditions (in μg/cm-2/h)”. Because of very low permeated amounts we decided to use ng·cm-2·h-1 instead of µg·cm-2·h-1 in case of H-DMCA.
- lines 412-414 There is a conceptual error in the presentation of the results. The flux is obtained from the slope of the straight line passing through the points ... the numerical data that the authors report are not related to the flux but of permeated amount per cm2 after a certain time interval. Authors need to reconsider their deductions
According to the OECD guidance document (reference [11]) flux is defined as “mass of test substance passing through a unit area of skin per unit of time under steady-state conditions (in μg/cm2/h)”. We calculated the flux values by dividing the resorbed amounts [µg] after the respective time through the skin area [cm²] and the respective time [h]. Therefore, we consider that the reported numerical data are in accordance with the OECD definition of flux and that we can keep our deductions.
ALL ALONG THE MANUSCRIPT:
- please avoid the use of the personal form.
We changed the respective parts.
We thank the reviewer for the efforts to help us improve our manuscript.
Round 2
Reviewer 1 Report
Author addressed all my comments.
Author Response
Reviewer 1: Author addressed all my comments.
We thank the reviewer for the time and effort.

Reviewer 2 Report
ABSTRACT:
- Lines 15, 18, 24: please avoid the use of the personal form.
The personal form is no longer used.
Ok
- please insert the TEWL values for the employed skin samples
We inserted the TEWL values.
Ok
- line 22: does “absorbed into the skin” mean “accumulated into the tissue”? Generally the absorption process refers to active entry into the bloodstream and thus is should be better to talk about tissue accumulation. Moreover, it is not clear the timepoint at which the “absorption” data are related and these results sound strange together with the previously reported percentage of permeated actives.
Information on the timepoint (48 h) was added. In the OECD guideline it is written that “the test substance remaining in the skin should be considered as absorbed“. We also defined the different steps of absorption in Lines 320-324. For clarity in the abstract, the term “absorbed into the skin” was changed to “absorbed (penetrated into the skin)” (Line 24) and in Line 321 “tissue accumulation” was added for the skin penetration step.
Ok
- I suggest to insert the acronym of sesquiterpene lactones (STLs) also in the abstract to avoid long repetition and make it more fluid.
The acronym of sesquiterpene lactones (STLs) was inserted in the abstract.
Ok
MATERIALS AND METHODS:
- please specify the number of repetition for the permeation assays.
The number of used skin punches (technical replicates) was added.
Ok
- Line 89-91: Authors should better specify what they are ID 347400. ID 1732 and ID 1728. Are they ethical approval codes?
The IDs are internal donor IDs. We changed the description to “internal donor ID” for clarity. Ethical review and approval were not required for this study because only ex vivo materials were used that came from plastic surgery with the patient’s consent or directly from the breeder (Ellegaard Göttingen Minipigs A/S, Dalmose, Denmark). We added this information in the Materials Section.
Ok, I strongly suggest adding into the manuscript this above-description about that the specimens used for the experiments don’t require the Ethical review and approval.
- line 111: the name and manufacturer of the coating thickness gauge is missing, please add it.
The name and manufacturer of the coating thickness gauge were added.
Ok
- Line 112: the type (e.g. Franz diffusion cell), characteristics (eg. the volume of both receptor and donor chamber), name and manufacturer of diffusion cell are missing, please add them.
The volume of donor and acceptor chambers was added. The diffusion cells were in house custom-made, so no name and manufacturer can be provided. This information was added in the text.
Ok
- Lines 116-120: the volume both of the receptor chamber and the samples collected at each time points is missing, please improve this part.
At each time point the receptor fluid was removed in its entirety and replaced with new PBS so that skin samples were completely wetted. Because of different acceptor chamber sizes, the volumes differed slightly but were considered in the calculations of the results.
Ok
- Line 126: this sentence is confused, what the “5% acetonitrile” is referred to? Seems to be related to the solvent, but is pure acetonitrile or a mixture with other solvents?
“5% Acetonitrile” is referred the solvent which is acetonitrile in aqueous solution (5/95, v/v). We added (aqueous) to clarify the solvent mixture.
Ok, I suggest rewriting the sentence as “were dissolved in aqueous solution containing the 5% (v/v) of acetonitrile”.
RESULTS AND DISCUSSION:
- lines 213-214: TEWL values are reported without a SE. Moreover, as the experiments should be repeated it is unlikely to imagine that different skin samples have identical TEWL. Please clarify if the repetition were carried out by using different portion of the same tissue or different similar tissues.
TEWL values were determined in triplicates at different locations on the same piece of tissue. This information and the deviations were added in the Materials Section.
Ok
- Table 1: the calibration curves of all the respective STLs compound are missing, I suggest to add them in the supplementary material.
The calibration curves were added in the Supplementary Materials (Figures S4-S11).
Ok
- lines 216-220: this is a repetition of what previously described in the methods section.
The redundant technical information was removed from these sentences. However, we feel that it is necessary to keep the main information as a service for the reader.
Ok
- Table 2: at which timepoint does the table refer? 48 h? please specify.
The timepoint was added in the Table caption.
Ok
- Lines 225-227: this sentence is misleading, please improve it.
We improved the sentence by specifying the distribution of STLs.
Ok
- Line 322: I strongly suggest adding the description of this paragraph in the respective methods section.
A description of this paragraph was added in the respective Methods Section 2.4.
Ok
-figure 4 (1) the ordinate label is wrong, because the values are expressed as a percentage. I guess the authors intended the percentage of the dose .... This needs to be clarified. Also the ordinate label of figure 4 (1) is wrong… maybe it should be ng/cm2
In the Figure captions of Figures 2 and 4 “percentage of total applied dose” was added. The ordinate label of Figures 2 (2) and 4 (2) are µg·cm-2·h-1 (Arnica STLs) and ng·cm-2·h-1 (H-DMCA) because in the OECD guidance document (reference [11]) flux is defined as “mass of test substance passing through a unit area of skin per unit of time under steady-state conditions (in μg/cm-2/h)”. Because of very low permeated amounts we decided to use ng·cm-2·h-1 instead of µg·cm-2·h-1 in case of H-DMCA.
The ordinated labels are not fixed. Anyway, the ordinate label in figure 2 and 4 (2) is still wrong since the x-axis is referred to the time (h) so it should be correct that the ordinate label was ng/cm-2 and not ng/cm-2/h. Please review better this part.
- lines 412-414 There is a conceptual error in the presentation of the results. The flux is obtained from the slope of the straight line passing through the points ... the numerical data that the authors report are not related to the flux but of permeated amount per cm2 after a certain time interval. Authors need to reconsider their deductions
According to the OECD guidance document (reference [11]) flux is defined as “mass of test substance passing through a unit area of skin per unit of time under steady-state conditions (in μg/cm2/h)”. We calculated the flux values by dividing the resorbed amounts [µg] after the respective time through the skin area [cm²] and the respective time [h]. Therefore, we consider that the reported numerical data are in accordance with the OECD definition of flux and that we can keep our deductions.
The reference of the OECD is correct, but authors mislead the meaning of steady state. Indeed, at the steady state conditions the flux value through the skin is the slope of the straight line passing through the point before the plateaux occurred. Hence, it is not correct to evaluate the flux at each time point. The graphs should be prepared again by plotting the amount of drug permeated per unit area as a function of time. Then the linear portion should be considered to calculate the fluxes. See some references as https://doi.org/10.3390/pharmaceutics13091370.
The interpretation of the data in terms of flux is crucial to define the validity of the research, however the authors still confuse the quantities of drug at each time point with the flux.
ALL ALONG THE MANUSCRIPT:
- please avoid the use of the personal form.
We changed the respective parts.
Ok
We thank the reviewer for the efforts to help us improve our manuscript.
Author Response
- Line 89-91: Authors should better specify what they are ID 347400. ID 1732 and ID 1728. Are they ethical approval codes?
The IDs are internal donor Ids. We changed the description to “internal donor ID” for clarity. Ethical review and approval were not required for this study because only ex vivo materials were used that came from plastic surgery with the patient’s consent or directly from the breeder (Ellegaard Göttingen Minipigs A/S, Dalmose, Denmark). We added this information in the Materials Section.
Ok, I strongly suggest adding into the manuscript this above-description about that the specimens used ort he experiments don’t require the Ethical review and approval.
We added this information in Lines 98-101.
- Line 126: this sentence is confused, what the “5% acetonitrile” is referred to? Seems to be related to the solvent, but is pure acetonitrile or a mixture with other solvents?
“5% Acetonitrile” is referred the solvent which is acetonitrile in aqueous solution (5/95, v/v). We added (aqueous) to clarify the solvent mixture.
Ok, I suggest rewriting the sentence as “were dissolved in aqueous solution containing the 5% (v/v) of acetonitrile”.
We changed the formulation to ”were dissolved in water/ acetonitrile (95/5, v/v)“.
-figure 4 (1) the ordinate label is wrong, because the values are expressed as a percentage. I guess the authors intended the percentage of the dose .... This needs to be clarified. Also the ordinate label of figure 4 (1) is wrong… maybe it should be ng/cm2
In the Figure captions of Figures 2 and 4 “percentage of total applied dose” was added. The ordinate label of Figures 2 (2) and 4 (2) are µg·cm-2·h-1 (Arnica STLs) and ng·cm-2·h-1 (H-DMCA) because in the OECD guidance document (reference [11]) flux is defined as “mass of test substance passing through a unit area of skin per unit of time under steady-state conditions (in μg/cm-2/h)”. Because of very low permeated amounts we decided to use ng·cm-2·h-1 instead of µg·cm-2·h-1 in case of H-DMCA.
The ordinated labels are not fixed. Anyway, the ordinate label in figure 2 and 4 (2) is still wrong since the x-axis is referred to the time (h) so it should be correct that the ordinate label was ng/cm-2 and not ng/cm-2/h. Please review better this part.
We prepared the graphs again by plotting the amount of drug permeated per unit area as a function of time. In the new graphs the ordinate labels are µg/cm² and ng/cm2 as requested. After this modification, the two graphs for each skin sample could be united in one diagram with two different ordinate axes on the left and right side.
- lines 412-414 There is a conceptual error in the presentation of the results. The flux is obtained from the slope of the straight line passing through the points ... the numerical data that the authors report are not related to the flux but of permeated amount per cm2 after a certain time interval. Authors need to reconsider their deductions
According to the OECD guidance document (reference [11]) flux is defined as “mass of test substance passing through a unit area of skin per unit of time under steady-state conditions (in μg/cm2/h)”. We calculated the flux values by dividing the resorbed amounts [µg] after the respective time through the skin area [cm²] and the respective time [h]. Therefore, we consider that the reported numerical data are in accordance with the OECD definition of flux and that we can keep our deductions.
The reference of the OECD is correct, but authors mislead the meaning of steady state. Indeed, at the steady state conditions the flux value through the skin is the slope of the straight line passing through the point before the plateaux occurred. Hence, it is not correct to evaluate the flux at each time point. The graphs should be prepared again by plotting the amount of drug permeated per unit area as a function of time. Then the linear portion should be considered to calculate the fluxes. See some references as https://doi.org/10.3390/pharmaceutics13091370 .
The interpretation of the data in terms of flux is crucial to define the validity of the research, however the authors still confuse the quantities of drug at each time point with the flux.
We prepared the graphs again by plotting the amount of drug permeated per unit area as a function of time. Then, we considered the linear portion to create a regression line, whose slope then yielded the steady-state fluxes. In case of porcine skin and human skin A we considered the linear part between 24 h and 48 h, for human skin B between 4 h and 32 h and for H-DMCA between 8 h and 48 h. We added the corrected steady-state flux values (lines 328-330 and 463).
We thank the reviewer for the scrutiny and are very grateful for the improvement of our work.
